# Soil Bacterial Community and Greenhouse Gas Emissions as Responded to the Coupled Application of Nitrogen Fertilizer and Microbial Decomposing Inoculants in Wheat (*Triticum aestivum* L.) Seedling Stage under Different Water Regimes

Djifa Fidele Kpalari [1,2,3], Abdoul Kader Mounkaila Hamani [4,*], Cao Hui [1,2], Jean Mianikpo Sogbedji [3], Junming Liu [1,2], Yang Le [1,2], Rakhwe Kama [5] and Yang Gao [1,*]

[1] Key Laboratory of Crop Water Use and Regulation, Ministry of Agriculture and Rural Affairs, Institute of Farmland Irrigation of CAAS, Xinxiang 453002, China; 2021y90200001@caas.cn (D.F.K.); 82101221063@caas.cn (C.H.); 82101211059@caas.cn (J.L.); 82101212135@caas.cn (Y.L.)

[2] Graduate School of Chinese Academy of Agricultural Sciences, Beijing 100081, China

[3] Laboratoire Interface Sciences du Sol, Climat et Production Végétale (LISSCPV), Ecole Supérieure d'Agronomie, Université de Lomé, Lome 01 BP 1515, Togo; jsogbedji@ifdc.org

[4] College of Tropical Crops, Hainan University, Haikou 570100, China

[5] College of Natural Resources and Environment, South China Agricultural University, Guangzhou 510642, China; 2020y90100008@caas.cn

* Correspondence: 2020y90100004@caas.cn (A.K.M.H.); gaoyang@caas.cn (Y.G.); Tel./Fax: +86-15652472605 (A.K.M.H.); +86-373-3393224 (Y.G.)

**Abstract:** The soil microbial community is critically important in plant nutrition and health. However, this community is extremely sensitive to various environmental conditions. A pot experiment was conducted during the wheat seedling stage to better understand the influences of the coupled application of nitrogen (N) and microbial decomposing inoculants (MDI) on the soil bacteria community under different water regimes. There were two levels of water and six levels of fertilization. The results reveal that water stress increased the relative abundance of *Acidobacteria* and decreased that of *Firmicutes* and *Proteobacteria*. The application of 250 kg N ha$^{-1}$ altered the diversity of the bacterial community but increased the relative abundance of nitrifying bacteria. Nitrous oxide ($N_2O$) and carbon dioxide ($CO_2$) emissions were negatively correlated with *Myxococcota* and *Methylomirabilota* while positively correlated with *Patescibacteria*. These two gases were also positively correlated with nitrifying bacteria, and the correlation was more significant under the full irrigation regime. These findings indicate that MDI does not substantially influence the soil bacterial community and its relationship with greenhouse gas emission at the wheat seedling stage and that the abundance of the soil bacterial community would mainly depend on the rational control of the amount of N and water applied.

**Keywords:** water regimes; nitrogen fertilization; microbial decomposing inoculants; soil bacterial community; greenhouse gas emissions

## 1. Introduction

Drought is a natural disaster that can affect all aspects of life. It is a very complex climatic phenomenon that has yet to be fully understood [1,2]. Statistical data show that drought affected over 25% of the world's population and induced an overall economic loss of $175.10^{12}$ dollars between 1900 and 2019 [3]. According to predictions, such climatic events will continue to occur more frequently in the coming years [4,5]. The agricultural sector is one of the most vulnerable to water deficits [6]. Indeed, water is an essential resource for crop production [7–9], and its absence or deficit leads to considerable yield losses. The yield loss caused by drought annually exceeds that caused by all crop pathogens [10].

According to Ullah, et al. [11], water is the second limiting factor in crop production after soil. For Hussain, et al. [12], yield losses induced by water deficit can range from 30 to 92%, depending on the considered crop. However, the sensitivity of yield to this climatic event mainly depends on the type of agricultural product harvested (root, leaf, grain, etc.) [13]. This climatic stress also deteriorates the quality of cereal products, such as winter wheat, by reducing the size of the grains and their organic matter content [14]. Drought negatively affects crop production by inducing considerable changes in the plant environment, particularly by negatively influencing soil nutrient availability [15].

Among the essential nutrients for plant growth and development, nitrogen (N) appears to be the most limiting nutrient in crop production. This nutrient plays a crucial role in the various biochemical reactions within the plant, including photosynthesis [16,17], which makes it a determining factor in yield formation [18,19]. The plant's efficiency of nitrogen use (NUE) depends on its application technique. For example, split N application improves physiological characteristics and yield [20,21], and the effectiveness of this technique increases as input timing becomes optimal [22]. Deep N application improves NUE and crop yield while reducing ammonium N losses [23–25]. Previous studies on the combined application of N fertilizers and microbial decomposing inoculants (MDI) have reported improved crop growth and productivity with reduced greenhouse gas (GHG) emission [26–28]. These different N supply techniques benefit plants only under favorable climatic conditions.

Water and N are essential for adequate yield formation [29,30]. Unavailability or low soil moisture significantly affects the availability of different forms of N in the soil [31]. Previous studies have shown that drought inhibits soil N ammonization and nitrification reactions [32,33] but decreases soil N loss [34]. This climatic stress impacts the biogeochemical processes of nitrogen in the soil by negatively influencing the biological agents involved in the biogeochemical cycle of nutrients in the soil, as well as the rate of GHG emissions.

Soil microbial organisms play an essential role in forming and protecting soil quality. These organisms are present in negligible quantities in the soil but ensure soil nutrient mineralization [35–37]. Several studies have reported the central role of soil microorganisms in the mineralization of organic matter [38–40], nitrogen [41–43] and phosphorus in the soil [44,45]. Nevertheless, soil microbes are susceptible to different environmental stresses. Several studies have been conducted to demonstrate the impact of drought on the soil microbial community. It has been proven that drought significantly alters the structure, size, and composition of the soil microbial community, leading to undesirable changes in the biogeochemical cycling of soil nutrients [46,47]. Xu, et al. [48] have shown that drought leads to a 17% drop in microbial biomass, whereas high precipitation leads to an 18% increase. Moreover, the influence of water stress is more pronounced on bacteria than fungi [47,49,50]. Siebielec, et al. [51] have reported significant changes in the relative abundance of different soil bacterial phyla under water deficit conditions. Other factors, including soil fertilization, also influence the response of the soil microbial community to drought. It is currently known that an optimal application of N fertilizer mitigates the impact of drought on the microbial community of the soil [52,53]; meanwhile, as its excessive application increases in the long term, so does the sensitivity of the microbial community to various environmental stresses [54]. Other studies have reported a significant improvement in microbial biomass and an alteration in the composition of the soil microbial community under suitable N fertilization in drought conditions [55,56]. Despite these numerous previous studies, the combined effect of biological and mineral fertilizers on the soil microbial community under water deficit conditions remains less clear.

The soil microbial community contributes in various ways to the GHG emissions, and their contribution varies depending on the stresses to which they are subjected. For example, soil moisture, salinity, and N availability are all factors that influence the relationship between soil microbes and GHG emissions [57–59]. In addition, the different stages of plant development influence in various ways the soil microbial community [60–62]. However, very little research has taken into account these different stages until now. In fact, the



seedling stage is very critical for the proper development of wheat and is sensitive to various environmental stresses [63–66]. The present study aims to characterize the soil microbial community at the seedling stage of winter wheat and its contribution to greenhouse gas emissions under the combined application of biological and mineral fertilizers. We hypothesize that applying *bacillus*-based microbial inoculants combined with a high dose of nitrogen would increase the abundance of bacillus and soil-nitrifying bacteria under full irrigation and reduce their contribution to GHG emissions. This hypothesis is based on previous work, which has shown that, under the proper irrigation regime, the application of MDI attenuates the emission of GHG [27], whereas the opposite effect followed by an increase in the abundance of nitrifying bacteria is observed with the excessive application of nitrogen fertilizer [67,68].

## 2. Materials and Methods

### 2.1. Description of the Site and Experimental Design

The study was conducted in a controlled greenhouse at the Xinxiang Comprehensive Experimental Station of the Chinese Academy of Agricultural Sciences (35.09° N, 113.48° E, and altitude 81 m). The humidity inside the greenhouse was maintained between 40–50%, the photoperiod at 12 h, and the ambient temperature between 30 °C and 20 °C, day and night, respectively. Seeds were sown in 6.5 kg air-dried sandy loam soil (USDA-NRCS Soil survey division) contained in 5.3 L pots with 15 cm diameter and 30 cm height. The experimental soil's sand, silt, and clay averaged 55.59, 40.39, and 4.11%, respectively. The average soil bulk density was 1.52 g cm$^{-3}$. The field capacity (31.62%) and the average value for soil permanent wilting point was (17.34%). The soil's average values of available soil N, phosphorous and potassium were 43.77, 16.43 and 129.83 mg kg$^{-1}$, respectively. The average values of soil electric conductivity, pH, and organic matter were 142.87 s cm$^{-1}$, 8.59, and 1.01%, respectively. Seedlings were trimmed to four seedlings per pot after ten days post germination. Two-factorial design with six replicates per treatment was adopted for the experiment implementation. The first factor included two irrigation levels, and the second included six fertilization levels. A total of 12 treatments were defined for this study (Table 1).

**Table 1.** Description of the experimental treatments.

| Treatments | Irrigation Regime (%FC) | Nitrogen Rate (%) | MDI |
|---|---|---|---|
| R1N1 | 80 | 0 | No |
| R1N2 | 80 | 0 | Yes |
| R1N3 | 80 | 100 | Yes |
| R1N4 | 80 | 50 | Yes |
| R1N5 | 80 | 100 | No |
| R1N6 | 80 | 50 | No |
| R2N1 | 50 | 0 | No |
| R2N2 | 50 | 0 | Yes |
| R2N3 | 50 | 100 | Yes |
| R2N4 | 50 | 50 | Yes |
| R2N5 | 50 | 100 | No |
| R2N6 | 50 | 50 | No |

Note: FC = field capacity; MDI = microbial decomposing inoculants.

The two irrigation levels were, respectively, 80% and 50% of the field capacity of the soil. Based on the soil's dry weight in each pot and the field capacity, soil moisture was determined with daily changes in the pot's weight, weighed with an electronic balance at 8:00 a.m. Regarding fertilization, different doses of N (urea) were combined with MDI and applied to crops by fertigation. The local dose of N fertilization of wheat or maize in northern China, set at 250 kg N ha$^{-1}$ or 100 mg kg$^{-1}$, was considered to estimate the maximum amount of N to be applied [69]. Using urea (46% N), 217 mg kg$^{-1}$ was applied for plants receiving 100% N and 108 mg kg$^{-1}$ for plants receiving 50% N. BioTech

Medics Inc (BMCS) Co., Ltd. (Tokyo, Japan) supplied the MDI. This inoculant with high N fixing capacities is composed of *Bacillus brevis* ($1.6 \times 10^9$ cells $g^{-1}$), *Bacillus laterosporus* ($3.2 \times 10^9$ cells $g^{-1}$), and *Saccharomyces* ($5.2 \times 10^9$ cells $g^{-1}$), and was applied after dilution at a concentration of 1 L of inoculant for 50 L of water. The Zhoumai22 winter wheat variety served as plant material for the experiment.

### 2.2. Soil Sampling and Measurement of Inorganic Nitrogen and GHG Emissions

Soil samples were collected between 0–10 cm soil depth under each treatment using an auger. A first sample was used to measure the soil's $NH_4$ and $NO_3$. A second sample (collected in the root environment) was stored in liquid nitrogen just after sampling, transferred to the laboratory, and then stored again at a temperature of $-80\ °C$ for the extraction of DNA. The soil $NH_4$ and $NO_3$ concentrations were measured by continuous flow analyzer as described by Ning, et al. [70]. GHG ($N_2O$ and $CO_2$) emissions were measured by the static chamber method [71]. For this, three plants were randomly selected under each treatment for measurement, and Li-6400-01 Liquefaction small steel bottle was used for gas collection [27]. The collection of soil samples for chemical and microbial analysis and the GHS measurements were carried out on the day of harvesting (46th day after germination).

### 2.3. DNA Extraction and Sequencing

DNA extraction was performed in 0.5 g of soil using E.Z.N.A Soil DNA Kit (Omega Bio-tek, Norcross, GA, USA), and purity was assessed using ultraviolet–visible spectrophotometer (Thermo Scientific, Wilmington, NC, USA). DNA quality was checked by 1% agarose gel electrophoresis. The hypervariable region V3-V4 of the bacterial 16S rRNA was amplified using specific primers (338F: 5′-ACTCCTACGGGAGGCAGCAG-3′; 806R: 5′-GGACTACHVGGGTWTCTAAT-3′) (GeneAmp 9700, ABI, Cambridge, MA, USA). The PCR amplification was performed using the following process: 27 cycles of denaturation at 95 °C for 30 s, annealing at 55 °C for 30 s, extension at 72 °C for 30 s, and final extension at 72 °C for 10 min. The PCR mixtures contained 4 μL of $5 \times$ TransStart FastPfu buffer, 2 μL of 2.5-mM deoxynucleoside triphosphates (dNTP), 0.8 μL of forward primer (5 μM), reverse primer (5 μM) 0.8 μL, 0.4 μL of TransStart FastPfu DNA Polymerase, and 10 ng of template DNA. The PCR products were extracted from a 2% agarose gel and quantified using a Quantus Fluorometer system (Madison, WI, USA). Purified amplicons were pooled in equimolar and were paired-end sequenced on a MiSeq platform (Illumina, San Diego, CA, USA) by Majorbio Bio-Pharm Technology Co., Ltd. (Shanghai, China).

### 2.4. Statistical Analysis

Three replicates per treatment were used for the statistical analysis. The alpha diversity indices of each replicate were calculated using the online Majorbio Cloud Platform, www.majorbio.com (accessed on 10 July 2023), and one-way ANOVA allowed us to discriminate the averages of the indices at the 5% threshold using R software (4.3.1). Ordination Bray–Curtis distance was performed to visualize the dissimilarity using principal coordinate analysis (PCoA) with the Vegan package contained in R [72]. The relationships between the different environmental parameters and the soil microbial abundance were determined using the Pearson correlation. All graphics were made using R software.

## 3. Results

### 3.1. Bacterial Community Diversity

The diversity of a soil's microbial community is an essential parameter for assessing its biological and functional state. In the present study, the alpha diversity within bacterial communities and the dissimilarity between these communities were tested. Figure 1 shows the variation of the Shannon, Ace, and observed species number indices under the various water and fertilization treatments. The fertilization modes significantly influenced the diversity indices rather than the water regime.

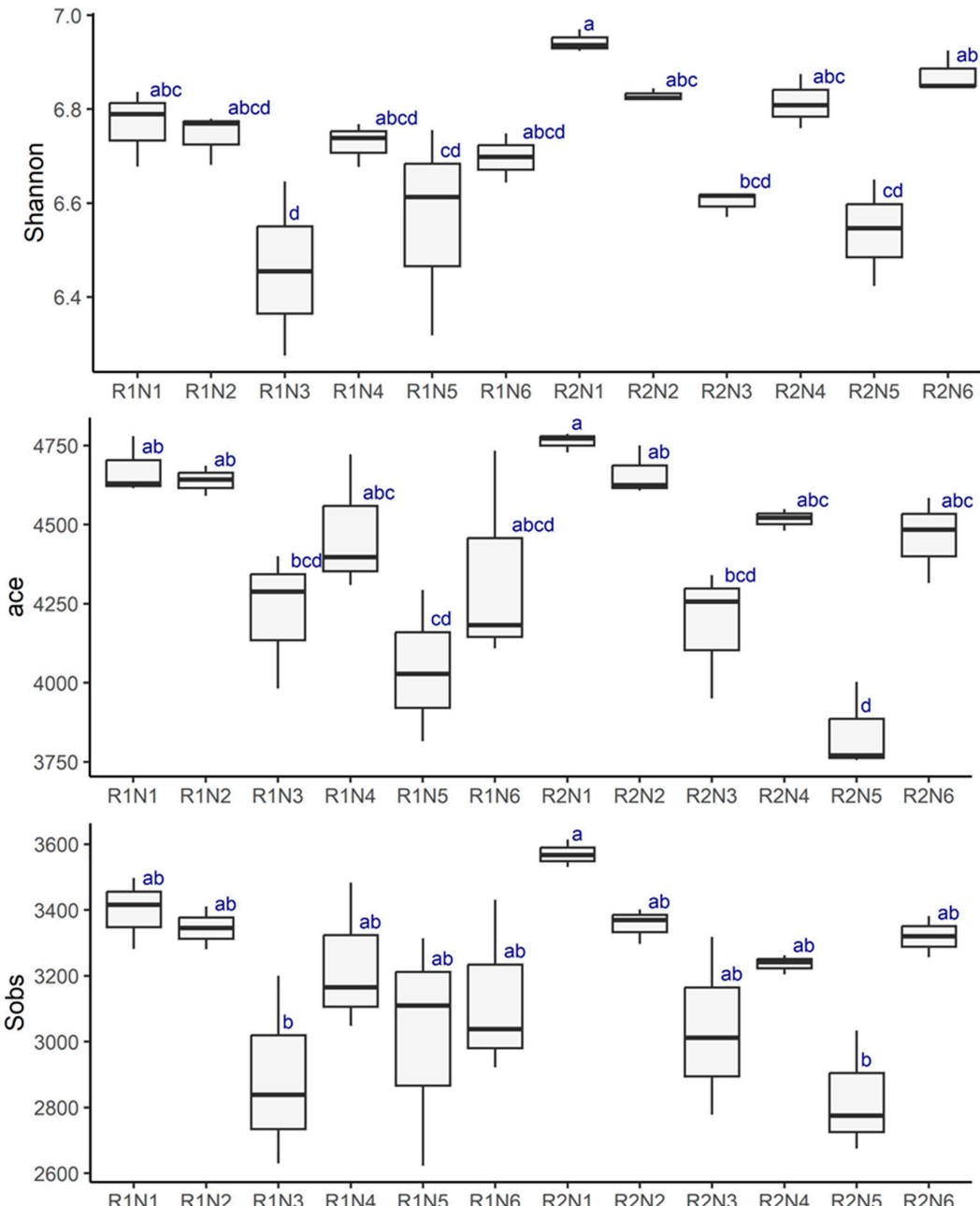

**Figure 1.** Alpha diversity of the soil bacterial community as affected by different fertilization strategies and water regimes. Box plots are shown representing the median and quartiles of the data. Data points with different alphabets are significantly different.

The high values of alpha diversity indices were obtained with the N1, N2, N4, and N6 fertilization modes under the two water regimes and their maximum values under R2N1. Low values of these indices were obtained under treatments with N3 and N5 fertilization modes under both water regimes. Alpha diversity was, therefore, high under the treatments without N or with a 50% N supply and low under the treatments with a 100% N supply.

Principal coordinate analysis (PCoA) was used to compare the distance between the bacterial communities of the different treatments (Figure 2). The two axes of each subfigure explain more than 50% of the dissimilarity between the bacterial communities subjected to various fertilization modes under each of the water regimes. The communities under the treatments N1, N2, N4, and N6 were relatively similar and differed from those under N3 and N5 in both water regime conditions. A similar trend was observed for alpha diversity.

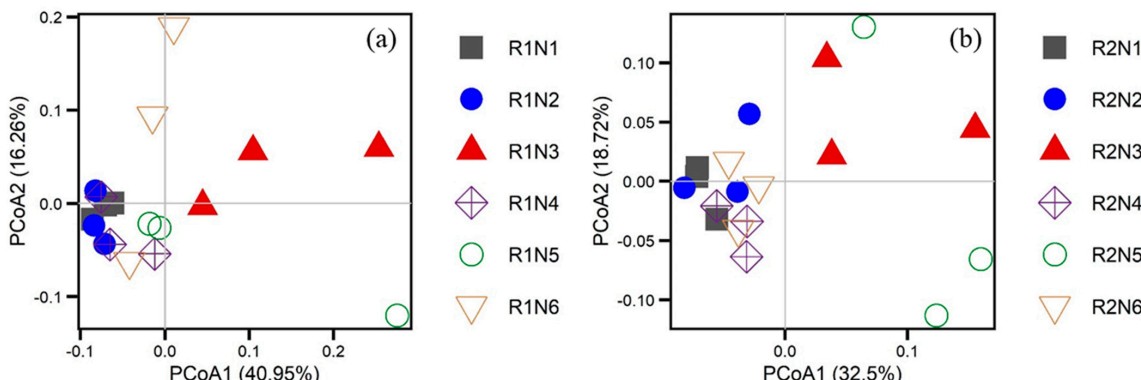

**Figure 2.** Principal coordinate analysis plot of Bray–Curtis among the different fertilization strategies under a full water regime (**a**) and a deficit water regime (**b**). The same color points represent the repetitions of the same fertilization strategies. The closer the points are, the greater the similarity between the bacterial communities they represent.

*3.2. Abundance and Composition of the Bacterial Community*

At the phylum level, *Proteobacteria* (22–33%), *Actinobacteria* (22–25%), *Acidobacteria* (9–22%), *Chroroflexi* (9–12%), and *Firmicutes* (2–7%) were the five most dominant phyla under all treatments and represent more than 80% of all microbial taxa (Figure 3). A low relative abundance of *Firmicutes* was observed under treatments with water deficit. *Acidobacteria*, unlike *Proteobacteria*, were more abundant under all treatments with water deficit and full water regime without N supply. An exceptionally high proportion of *Proteobacteria* and a low proportion of *Acidobacteria* were observed under R1N3. Although not among the most abundant phyla under all treatments, relatively low proportions of *Myxococcota* and *Methylomirabilota* and high proportions of *Pastescibacteria* were observed under treatments with N3, N4, and N5 from both water regimes.

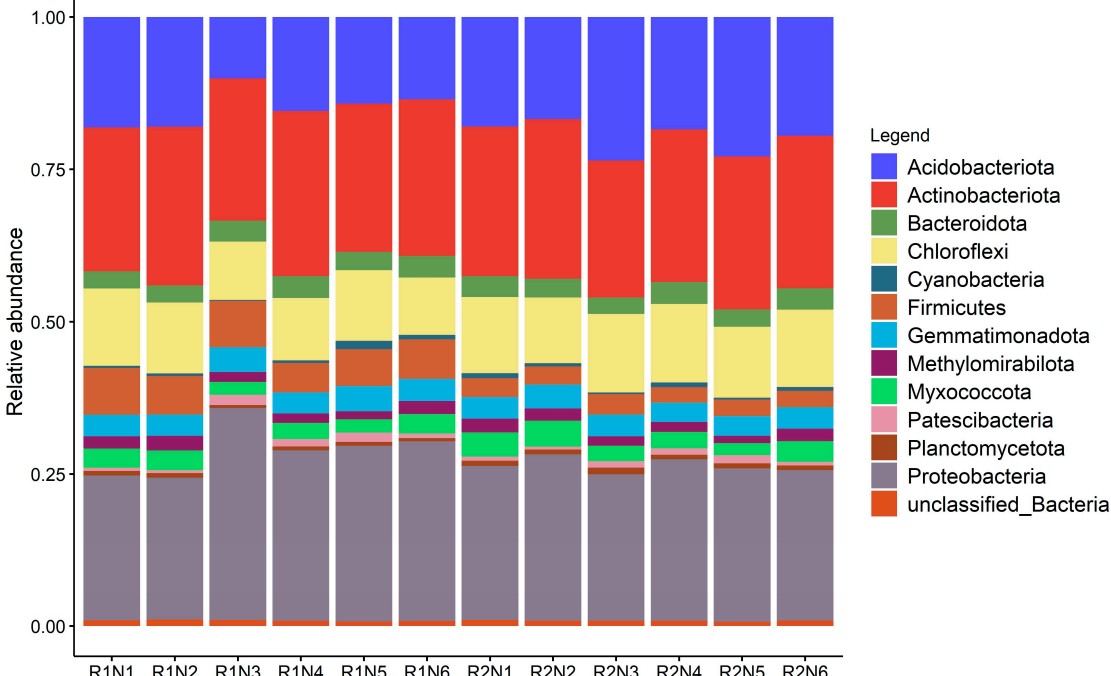

**Figure 3.** Relative abundance of soil bacteria communities at the phylum level according to six fertilization strategies and two water regimes. Relative abundance was calculated by averaging the abundances of replicate samples.

At the genus level, high proportions of *Vicinamibacteraceae*, *Vicinamibacterales*, *RB41*, and low proportions of *Bacillus* were obtained under the deficit irrigation regime. The proportion of *Sphingomonas* and *Bacillus* was exceptionally high under R1N3, while that of *Vicinamibacteraceae* and *Vicinamibacterales* was the lowest under this treatment (Figure 4). Compared with the control treatment, MDI had no significant influence on the soil microbial community.

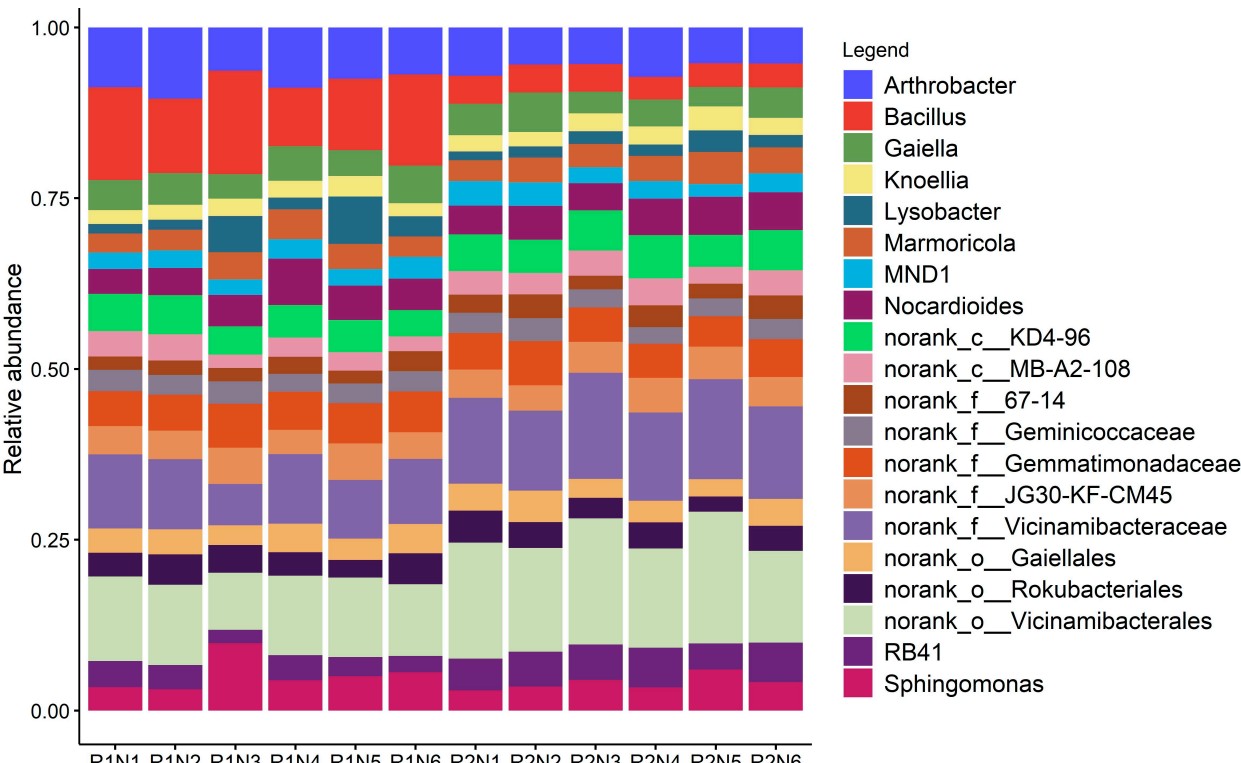

**Figure 4.** Relative abundances of the top 20 genera of soil bacteria according to six fertilization strategies and two water regimes. Relative abundance was calculated by averaging the abundances of replicate samples.

*3.3. Correlation Analysis between Bacteria Community and Environmental Parameters*

Under the two irrigation regimes, the concentrations of soil nitrate ($NO_3$) and ammonium ($NH_4$), as well as of nitrous oxide ($N_2O$) and carbon dioxide ($CO_2$), were high under the treatments, with 100% N application rates as compared with other treatments [27]. The correlation between the relative abundance at the phylum level of bacteria and environmental parameters is shown in Figure 5. Regardless of the irrigation regime and the fertilization strategies, $NO_3$ and $NH_4$ on one side and $N_2O$ and $CO_2$ on the other side were correlated in a substantially similar way with the different soil microorganisms. $N_2O$ and $CO_2$ were negatively correlated with *Myxococcota* and *Methylomirabilota* and positively correlated with *Patescibacteria* (Figure 5).

Under the two water regimes, the correlation of $NH_4$ and $NO_3$ with the different phyla of bacteria was insignificant. In contrast, a significant positive correlation was observed between *Acidobacteria*, and $N_2O$ and $CO_2$ under water stress. *Bacteroidota* and *Patescibacteria* were positively correlated with $NH_4$ and $NO_3$, respectively, under the treatments with the microbial inoculant. However, only *Patescibacteria* was significant and positively correlated with available forms of N under all fertilization strategies (Figure 5).

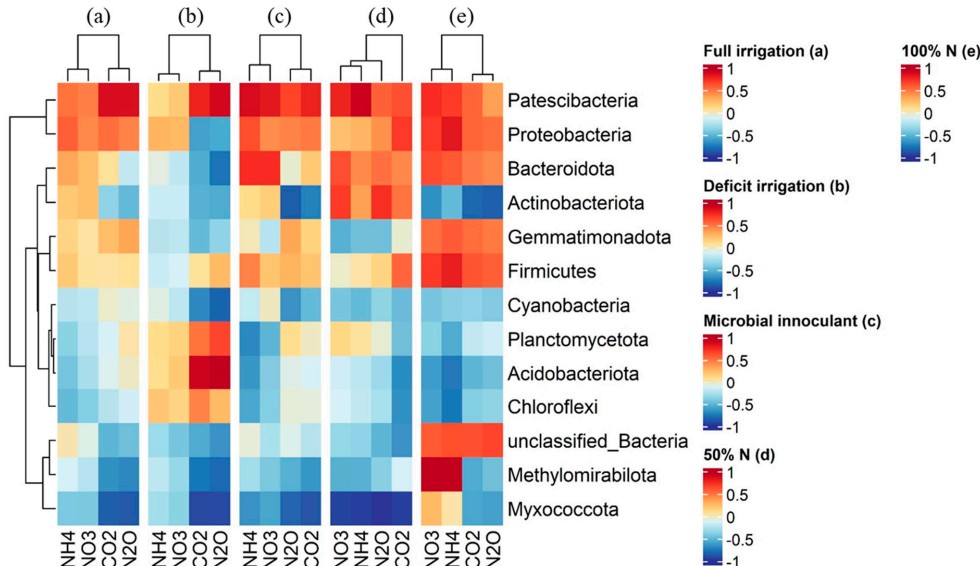

**Figure 5.** Pearson's correlation between environmental parameters and bacterial abundance at the phylum level according to full water regime (**a**), deficit water regime (**b**), treatments with microbial decomposing inoculants (**c**), treatments with 50% N fertilizer (**d**) and treatments with 100% N fertilizer (**e**).

### 3.4. Nitrifying Bacteria and Their Relationship to Environmental Parameters

The relative abundance, according to the treatments, of nitrifying bacteria *Nitrolancea*, *Nitrosomonas*, *Nitrosospira*, and *Nitrospira* was evaluated in this study (Figure 6A). The different fertilization modes influenced the relative abundance of these bacteria more than the water regime. R1N3, R1N5, R2N3, and R2N5 treatments increased the relative abundance of nitrifying bacteria compared with the other treatments. A significant positive correlation was observed between nitrifying bacteria and GHG emissions, which was more significant under the full irrigation regime. The correlation of these bacteria with soil-available N forms was insignificant under the two water regimes (Figure 6B).

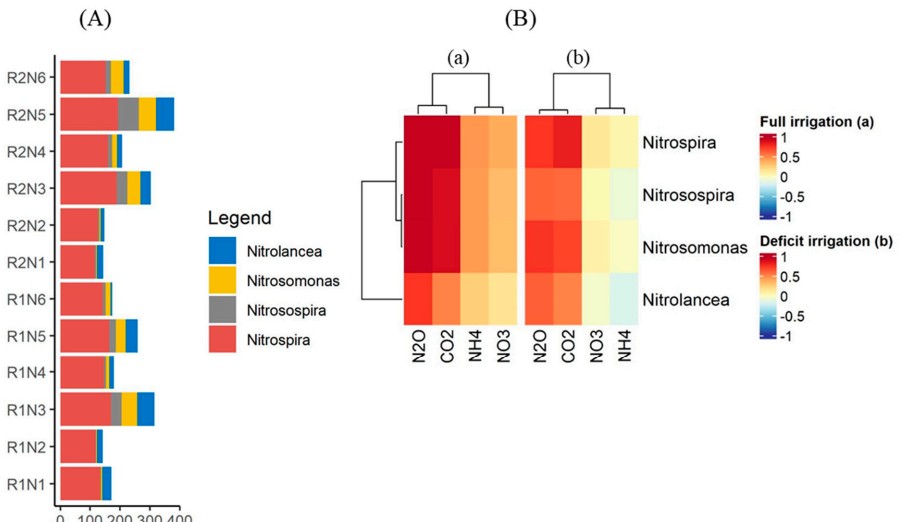

**Figure 6.** Nitrifying bacteria under different treatments. (**A**) Abundance of nitrifying bacteria according to the different fertilization strategies and water regimes. (**B**) Pearson's correlation between environmental parameters and nitrifying bacterial abundance according to full water regime (**a**) and deficit water regime (**b**).

## 4. Discussion

### 4.1. Influences of Various Water Regimes and Fertilization Modes on the Diversity of the Soil Bacterial Community

Knowing the factors influencing the diversity of the soil microbial community is essential for a better understanding of the changes within this community. Several factors, including soil texture, depth, acidity, and salinity, induce changes in the diversity of the soil microbial community [73–77]. In the present study, the alpha diversity and dissimilarity between the microbial communities under the different treatments varied much more depending on the dose of nitrogen supply rather than the irrigation regime and the application of the microbial decomposing inoculants. The application of the maximum nitrogen dose reduced the diversity (Figure 1) while increasing the distance between the bacterial communities of the different treatments (Figure 2). Previous studies have also reported the negative influence of nitrogen supply on the diversity of the soil microbial community [53,78,79]. Indeed, the application of nitrogen fertilizers composed of ammonia acidifies the soil [80,81], and the intensity of this acidity increases with the dose of nitrogen applied [82]. However, soil acidity is one of the factors that most influence the soil's microbial community. Thus, nitrogen fertilizers reduce soil microbial diversity by promoting increased soil acidity [83]. Wang, et al. [84] have shown that a drop in the microbial biomass of the soil accompanies this reduction in diversity under the application of nitrogen. Other studies have also reported the negative impact of reduced bacterial community diversity on soil health and nutrient cycling [85,86].

The detrimental impact of water stress on the diversity of the soil microbial community has been widely documented by previous studies [87–89]. In the present study, water deficit did not significantly influence the diversity of the soil bacterial community. This can be explained by the fact that the present study focused on the seedling stage, whereas most previous studies on drought have considered only the maturity stage of wheat. Several studies have shown the influence of different plant development stages on the response of the soil microbial community to different environmental stresses [90–93]. Therefore, future studies on drought need to consider the different stages of plant development to provide a clearer understanding of this climatic stress on the soil microbial community.

### 4.2. Effects of Different Water Regimes and Fertilization Modes on the Abundance of Various Soil Bacteria

*Bacillus* is a bacterial genus belonging to the *Firmicutes* phylum. The MDI used in this study is composed of 48% *Bacillus*. A small proportion of *Firmicutes*, particularly bacteria of the genus *Bacillus*, were observed under water stress conditions (Figures 3 and 4). In addition, the application of MDI or N fertilizer showed insignificant influence on the relative abundance of soil *Bacillus* under the different treatments (Figure 4). Previous studies have revealed that *Bacillus*-based microbial inoculants help mitigate the effects of drought on crop growth and productivity while increasing their disease resistance [94–97]. Other research has also reported the contribution of several species of bacteria of the genus *Bacillus* in the fixation of atmospheric N [98–100]. Nevertheless, the effect of *Bacillus*-based microbial inoculants application on the relative abundance of soil *Bacillus* remains less documented. The results of this study reveal that *Bacillus*-based microbial inoculants have no significant impact on the relative abundance of *Bacillus* in the soil. At the same time, water stress negatively influences its abundance in the soil.

*Acidobacteria* was mainly represented by bacteria of the genus *Vicinamibacteraceae*, *Vicinamibacterales*, and *RB41*, while *Proteobacteria* was represented by bacteria of the genus *Lysobacter* and *Sphingomonas* in the present study. The relative abundance of *Acidobacteria* was high, and that of *Proteobacteria* was low under water stress conditions and full irrigation without N supply (Figure 3). Previous studies have reported similar results [101,102]. Water stress compromises the biogeochemical cycle of various soil nutrients and hinders their bioavailability [103,104]. In addition, *Acidobacteria* generally abounds in oligotrophic and acidic environments [105–109], while *Proteobacteria* prefer copiotrophic environments with

an acidity relatively neutral or slightly alkaline [106,110–112]. This would explain the high proportion of *Acidobacteria* and low proportion of *Proteobacteria* observed under water stress conditions and a full irrigation regime without nitrogen supply. On the other hand, this would also explain the exceptionally high proportion of *Sphingomonas* (*Proteobacteria*), *Bacillus* (*Firmicutes*), and the meager proportions of *Vicinamibacteraceae* and *Vicinamibacterales* (*Acidobacteria*) observed under the R1N3 treatment (Figure 4).

### 4.3. Relationship between the Relative Abundance of Soil Microorganisms and Environmental Parameters

The contribution of soil microorganisms in the production of greenhouse gases remains a less documented subject. *Myxococcota*, *Methylomirabilota*, *Pastescibacteria*, and *Bacteroidota* were the bacterial phyla that mostly correlated with environmental parameters in this study (Figure 5). Results of previous work on the interaction between microbial abundance and GHG emissions remain divergent. Wang, et al. [113], who investigated biochar and microbial agent additives, observed that *Chloroflexi*, *Myxococcota*, *Acidobacteriota*, *Firmicutes*, and *Gemmatimonadota* are the phyla of bacteria that most influence GHG emissions. Other research on intermittent aeration during composting processes reports that *Proteobacteria*, *Chloroflexi*, *Bacteroidetes*, and *Actinobacteria* are the phyla most linked to GHG emission [114]. Thus, the interaction between GHG emissions and soil microbial community mainly depends on the factors studied. GHG emissions were negatively correlated with *Myxococcota* and *Methylomirabilota*, and positively correlated with *Pastescibacteria* (Figure 5). In addition to an increase in $N_2O$ and $CO_2$ emissions, there is a rise in the abundance of *Pastescibacteria* and a reduction in *Myxococcota* and *Methylomirabilota* under the treatments with the 100% N application rate (Figure 3). These observations indicate that *Myxococcota*, *Methylomirabilota*, and *Actinobacteria* would have mitigating effects on GHG emissions, unlike *Pastescibacteria*. Regardless of irrigation regimes, *Patescibacteria* and *Bacteroidetes* were positively correlated with soil inorganic N (Figure 5). This result agrees with the finding of Ren et al. [115], who showed that *Patescibacteria* and Bacteroidetes have a positive influence on the availability of soluble forms of N in the soil.

### 4.4. Nitrifying Bacteria and Their Relationship with Environmental Parameters

The soil microbial community ensures the oxidation of ammonia to nitrate. The relative abundances of nitrifying bacteria of the genera *Nitrolancea*, *Nitrosomonas*, *Nitrosospira*, and *Nitrospira*, following water regime and fertilization strategies, were evaluated in this study (Figure 6A). The results show no significant influence of the water regimes or the application of MDI on the relative abundance of nitrifying bacteria and only the treatments with 100% N application rate had positive influences on the abundance of these bacteria. These observations agree with previous work, which also report a positive influence of N application on the abundance of soil nitrifying bacteria [3,67,116,117]. This could be explained by the addition of nitrogenous fertilizers forcing the nitrifying bacteria to multiply to meet the growing need for nitrification of the available ammonia.

A positive correlation was observed between nitrifying bacteria and $N_2O$ and $CO_2$ emissions, which was more significant under a full irrigation regime (Figure 6B). Similar results have been obtained by Sabba, et al. [118]. Furthermore, the nitrification reaction is one of the primary sources of $N_2O$ emissions into the atmosphere [68,119] and are sensitive to soil moisture [120]. Under a full irrigation regime, the oxygen content of the soil decreases, which accelerates the processes of denitrification and the emission of $N_2O$ [121,122]. Additionally, more studies have reported a positive relationship between $N_2O$ and $CO_2$ emissions [123–125]. These previous works explain the significant positive interaction between nitrifying bacteria and the GHG emissions observed under the full irrigation regime in this study. Thus, the excessive application of nitrogen fertilizer under a full irrigation regime would increase GHG emissions by increasing the abundance of nitrifying bacteria in the soil.

## 5. Conclusions

The effect of different fertilization strategies and water regimes on the soil microbial community and its relationship with GHG emissions in the wheat seedling stage was explored in this study. The results show that application of the maximum dose of N (250 kg N ha$^{-1}$) with or without MDI reduced the diversity of the soil bacterial community while increasing the relative abundance of nitrifying bacteria. The application of MDI had no significant influence on the bacterial community and its contribution to the emission of GHG. Moreover, unlike *Pastescibacteria* and nitrifying bacteria, the phyla of *Myxococcota* and *Methylomirabilota* seemed to attenuate N$_2$O and CO$_2$ emissions. At the wheat seedling stage, maintaining the balance of bacterial community in the soil would intrinsically depend on rational control of N application rates and water levels to crops. Furthermore, research on the optimal dose of nitrogen capable of improving crop yield requires considering the protection of soil microorganisms.

**Author Contributions:** Conceptualization, A.K.M.H. and Y.G.; methodology, A.K.M.H. and Y.G.; software, D.F.K.; validation, D.F.K., A.K.M.H. and Y.G.; formal analysis, D.F.K.; investigation, D.F.K. and A.K.M.H.; resources, Y.G.; data curation, D.F.K.; writing—original draft preparation, D.F.K.; writing—review and editing, A.K.M.H., Y.G., C.H., J.M.S., J.L., Y.L. and R.K.; visualization, D.F.K.; supervision, Y.G.; project administration, Y.G.; funding acquisition, Y.G. All authors have read and agreed to the published version of the manuscript.

**Funding:** This research was funded by the National Natural Science Foundation of China (No. 51879267), the China Agriculture Research System of MOF and MARA (CARS-03–19), and the Agricultural Science and Technology Innovation Program (ASTIP).

**Data Availability Statement:** Data will be made available on request.

**Conflicts of Interest:** The authors declare no conflict of interest.

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
