# Peer review of "Soil Bacterial Community and Greenhouse Gas Emissions as Responded to the Coupled Application of Nitrogen Fertilizer and Microbial Decomposing Inoculants in Wheat (Triticum aestivum L.) Seedling Stage under Different Water Regimes"

_agronomy, doi:10.3390/agronomy13122950_

Round 1
Reviewer 1 Report
Comments and Suggestions for Authors
Very nice written paper, good job!

Author Response
We are thankful for your time and efforts in deep and thorough revising this manuscript. We have addressed all suggestions and comments in the revised manuscript accordingly.
Reviewer 2 Report
Comments and Suggestions for Authors
I read the manuscript with high interest. This work is very actual now, under the global warming and drought increasing. This topic is quite relevant in terms of soil productivity and agricultural sector that are vulnerable to water deficits. The manuscript is well organized and nicely written. The presented results are very important and interesting to readers. The manuscript is well illustrated and very clear.
I would like to thank you for making an important contribution to investigating such a relevant topic.
However, there are some issues to address to, before the manuscript could be accepted.
1. Line 94: Please, make a correct reference according to journal requirements
2. Lines 92-93: The comma after “ways” is extra, I suppose. Please, rephrase the sentence
3.Line 116: What does mean irrigation levels 80 and 50%? How did you reach it?
4. Line 117: What quantity of MDI did you use? And Lines 123-125: the same comment
5. Line 121: Please, decode an acronym “BMCS”
6. Line 129: What exact chemical parameters did you measure?
7. Line 134: Hamani et al. [25]
8. Lines 163-166: Please, start this paragraph with some introduction with 2-5 sentences. Do not start Results paragraph with sentences about figure 1.
9. Lines 223-224: X-axis doesn’t have a subscript
10. Lines 298-302: Please, add some information about influence of MDI on results obtained.
11. Line 361: What dose of N fertilizer was analyzed?
12. Lines 376-392: Please, add units for X-axis of figure A
Author Response
Response letter to reviewer
Note: The line number mentioned by the editor in the comments is kept the same, according to the manuscript, whereas the line number mentioned in the response for the corresponding comments line is the new line number of the modified version of the manuscript.
Reviewer 2
- Line 94: Please, make a correct reference according to journal requirements
Response: Thank you for your observation. The reference has been revised accordingly. Line 102
- Lines 92-93: The comma after “ways” is extra, I suppose. Please, rephrase the sentence
Response: Thank you for your suggestion. The authors revised it as recommended. Lines 103-104
3.Line 116: What does mean irrigation levels 80 and 50%? How did you reach it?
Response: Thank you for your comment. Based on the soil’s dry weight in each pot and the field capacity, soil moisture was determined with daily changes in the pot’s weight, weighed with an electronic balance at 8:00 a.m. Lines 136-138
- Line 117: What quantity of MDI did you use? And Lines 123-125: the same comment
Response : The soil water content was controlled based on the field capacity. The MDI was applied as irrigation water. Therefore, the volume of 1/50 diluted MDI water was based on the field capacity.
- Line 121: Please, decode an acronym “BMCS”
Response: Thank you for your comment. It was revised as: BioTech Medics Inc (BMCS) Co., Ltd. (Japan). Line 142
- Line 129: What exact chemical parameters did you measure?
Response: NH4 and NO3 were measured. We have tried to be more precise in the revised manuscript. Line 155
- Line 134: Hamani et al. [25]
Response: Revised accordingly!
- Lines 163-166: Please, start this paragraph with some introduction with 2-5 sentences. Do not start Results paragraph with sentences about figure 1.
Response: Thank you for your suggestion. We added some sentences at the beginning of the results section. 187-189
- Lines 223-224: X-axis doesn’t have a subscript
Response: We redrawn the figures to make them more accurate.
- Lines 298-302: Please, add some information about influence of MDI on results obtained.
Response: Your comments have been taken into account and we added some information about the influence of MDI on the results obtained.
- Line 361: What dose of N fertilizer was analyzed?
Response: We have analyzed 100% N and 50% N. The figure in this section has been redrawn and information has been added to make it easier to understand.
- Lines 376-392: Please, add units for X-axis of figure A
Response: The X-axis in Figure A shows the number of nitrifying bacteria per treatment. So, there is no exact unity that can be attributed to it.

Reviewer 3 Report
Comments and Suggestions for Authors
1. The abstract of the study is presented. Please provide a more detailed explanation of the biological significance of the observed alterations in bacterial abundance. What are the potential consequences of changes in Acidobacteria, Firmicutes, and Proteobacteria for plant health and nutrition? Please provide further details regarding the probable ecological ramifications resulting from these alterations.
2. The assertion that MDI (Methyl Diphenyl Diisocyanate) does not significantly influence the soil bacterial community warrants more elaboration. This analysis aims to elucidate the factors that may have contributed to the limited impact of MDI (Methyl Bromide Disinfection) and explore the potential ramifications for future research endeavors and agricultural practices.
3. The introductory section successfully establishes the significance of water and nitrogen in crop production and the susceptibility of the agricultural sector to drought. Nevertheless, it is advisable to incorporate a more overt transition that directs attention toward the primary objective of the research, namely, examining the effects of microbial inoculants and nitrogen on both the soil microbial population and greenhouse gas emissions.
4. In the introductory section, provide additional details regarding the rationale behind selecting the seedling stage of winter wheat as the focus of this investigation. The significance of this stage lies in its connection to the overarching objectives of the research, as well as its pivotal role in comprehending the intricacies of the soil microbial community and its impact on greenhouse gas emissions.
5. The introduction explicitly presents the hypothesis, wherein the anticipated impacts of utilizing bacillus-based microbial inoculants in conjunction with a substantial nitrogen dosage are outlined. However, it is advisable to include a concise justification for the hypothesis to provide readers with a clearer understanding of the study's objectives.
6. Incorporate more contemporary research findings into the introduction to enhance the contextual framework and emphasize the originality of the current study despite acknowledging prior investigations into the influence of drought on the soil microbial community.
7. To enhance comprehension for readers who may not be familiar with the subject matter, it is imperative to provide precise definitions for phrases such as "new fertigation approaches." This will improve the accessibility of the introduction to a wider range of individuals.
8. In terms of methodology, the description of the study site and experimental design is presented clearly and straightforwardly. Nevertheless, the data obtained from this study offers additional insights into the soil properties of the experimental location, including pH levels, organic matter composition, and nutrient concentrations. Including this further information would augment the readers' comprehension regarding the soil conditions being examined.
9. The information provided in Table 1, which describes the experimental treatments, is informative. Including a concise elucidation of the abbreviations utilized in the table, such as "FC" denoting field capacity and "MDI" representing microbial decomposing inoculants, would facilitate readers' comprehension and interpretation of the data.
10. Elucidate the underlying justification for selecting nitrogen (urea) doses in the various treatments, particularly emphasizing the regional nitrogen fertilization dosage for wheat or maize cultivation in northern China. A lucid elucidation will enhance the scientific foundation for the selected nitrogen application rates.
11. The procedure for the soil sample and the subsequent analysis are thoroughly discussed. Please provide further information regarding the soil chemical characteristics measured in the initial sample. Furthermore, it is recommended to provide a detailed description of the techniques employed for measuring continuous flow analyzers and determining greenhouse gas emissions. This will contribute to the overall transparency and clarity of the methodology employed in the study.
12. Could you perhaps provide additional details regarding the alpha diversity indices that are calculated utilising the Majorbio Cloud Platform? Furthermore, it is important to incorporate details on the significance level employed in the one-way analysis of variance (ANOVA) and provide further clarification regarding the specific environmental elements that were taken into account during the Pearson correlation study.
13. The elucidation of alpha diversity findings is effectively presented, emphasising the impact of fertilisation methods on diversity indices rather than the water regime. The finding that certain fertilisation strategies resulted in substantial alpha diversity under both water regimes is a significant observation with valuable implications.
14. The present discussion offers a comprehensive examination of the impacts of water regimes and fertilisation modalities on the soil bacterial community. The correlation between the observed findings and the extant body of literature is firmly established.
15. The investigation of the effects of water deficit on the variety of the soil bacterial population is highly informative. Nevertheless, exploring plausible explanations for the apparent absence of substantial impact from water scarcity in this research is advisable, particularly when compared to previous scholarly works.
16. The presentation of the discussion on many genera, including Bacillus, Acidobacteria, and Proteobacteria, under diverse situations is commendable. Nevertheless, it is worth exploring in more depth the potential ecological consequences of the reported alterations in these species on soil health and the nutrient cycling process.
17. The full examination of nitrifying bacteria and their correlation with greenhouse gas emissions is presented. Please provide further details on how these findings enhance our comprehension of the nitrogen cycle and emissions within agricultural systems.
18. It is important to highlight the congruity or disparity between the present findings and the extant body of literature. Examine any inconsistencies and possible factors contributing to them. This would offer a more comprehensive viewpoint of the impact of this work on wider scientific knowledge.
Comments on the Quality of English LanguageModerate editing of the English language required
Author Response
Response letter to reviewer
Note: The line number mentioned by the editor in the comments is kept the same, according to the manuscript, whereas the line number mentioned in the response for the corresponding comments line is the new line number of the modified version of the manuscript.
Reviewer 2
- The abstract of the study is presented. Please provide a more detailed explanation of the biological significance of the observed alterations in bacterial abundance. What are the potential consequences of changes in Acidobacteria, Firmicutes, and Proteobacteria for plant health and nutrition? Please provide further details regarding the probable ecological ramifications resulting from these alterations.
Response: We thank you for your comments. Our study characterized the soil bacterial community and its relationship with GHG emission under the combined application of MDI and different doses of nitrogen. Discussing the health and nutritional status of the plant in our work would, therefore, be out of our research’s scope. However, we will consider all these parameters in future related studies.
- The assertion that MDI (Methyl Diphenyl Diisocyanate) does not significantly influence the soil bacterial community warrants more elaboration. This analysis aims to elucidate the factors that may have contributed to the limited impact of MDI (Methyl Bromide Disinfection) and explore the potential ramifications for future research endeavors and agricultural practices.
Response: Thank you for taking the time to understand our work. Memeanwhile, in this research, MDI means Microbial Decomposing Inoculum.
- The introductory section successfully establishes the significance of water and nitrogen in crop production and the susceptibility of the agricultural sector to drought. Nevertheless, it is advisable to incorporate a more overt transition that directs attention toward the primary objective of the research, namely, examining the effects of microbial inoculants and nitrogen on both the soil microbial population and greenhouse gas emissions.
Response: Thank you for your suggestion. We revised our transition sentences as suggested.
- In the introductory section, provide additional details regarding the rationale behind selecting the seedling stage of winter wheat as the focus of this investigation. The significance of this stage lies in its connection to the overarching objectives of the research, as well as its pivotal role in comprehending the intricacies of the soil microbial community and its impact on greenhouse gas emissions.
Response: We added more information on the importance of the seedling stage of wheat to the introduction.
- The introduction explicitly presents the hypothesis, wherein the anticipated impacts of utilizing bacillus-based microbial inoculants in conjunction with a substantial nitrogen dosage are outlined. However, it is advisable to include a concise justification for the hypothesis to provide readers with a clearer understanding of the study's objectives.
Response: Our hypothesis are justified as suggested.
- Incorporate more contemporary research findings into the introduction to enhance the contextual framework and emphasize the originality of the current study despite acknowledging prior investigations into the influence of drought on the soil microbial community.
Response: More recent information were added in the introduction as recommebnded.
- To enhance comprehension for readers who may not be familiar with the subject matter, it is imperative to provide precise definitions for phrases such as "new fertigation approaches." This will improve the accessibility of the introduction to a wider range of individuals.
Response: The sentence was modified to better comprehend our study.
- In terms of methodology, the description of the study site and experimental design is presented clearly and straightforwardly. Nevertheless, the data obtained from this study offers additional insights into the soil properties of the experimental location, including pH levels, organic matter composition, and nutrient concentrations. Including this further information would augment the readers' comprehension regarding the soil conditions being examined.
Response: Thank you for your suggestion. Soil parameters were added to the description of the study site and experimental design.
- The information provided in Table 1, which describes the experimental treatments, is informative. Including a concise elucidation of the abbreviations utilized in the table, such as "FC" denoting field capacity and "MDI" representing microbial decomposing inoculants, would facilitate readers' comprehension and interpretation of the data.
Response: Thank you for your appreciation
- Elucidate the underlying justification for selecting nitrogen (urea) doses in the various treatments, particularly emphasizing the regional nitrogen fertilization dosage for wheat or maize cultivation in northern China. A lucid elucidation will enhance the scientific foundation for the selected nitrogen application rates.
Response: Thank you for your observation. Our research was based on previously published articles, which suggested some optional urea concentration in such experimental conditions. We have also included information about nitrogen fertilization in our area in our manuscript with supporting references.
- The procedure for the soil sample and the subsequent analysis are thoroughly discussed. Please provide further information regarding the soil chemical characteristics measured in the initial sample. Furthermore, it is recommended to provide a detailed description of the techniques employed for measuring continuous flow analyzers and determining greenhouse gas emissions. This will contribute to the overall transparency and clarity of the methodology employed in the study.
Response: We added more information about nitrogen analysis procedures and measuring GHG emissions.
- Could you perhaps provide additional details regarding the alpha diversity indices that are calculated utilising the Majorbio Cloud Platform? Furthermore, it is important to incorporate details on the significance level employed in the one-way analysis of variance (ANOVA) and provide further clarification regarding the specific environmental elements that were taken into account during the Pearson correlation study.
Response: We provided additional information on the significance level of ANOVA. As far as alpha diversity indices are concerned, these are statistical formulae that will take many pages to incorporate into an article. As a result, many researchers calculate them using software or laboratory websites without going into too much detail about the formula, including (Yu et al., 2019; Qiu et al., 2021; Zhang et al., 2022; Han et al., 2023). Please these references at the end of this document.
- The elucidation of alpha diversity findings is effectively presented, emphasising the impact of fertilisation methods on diversity indices rather than the water regime. The finding that certain fertilization strategies resulted in substantial alpha diversity under both water regimes is a significant observation with valuable implications.
Response: Thank you for your appreciation
- The present discussion offers a comprehensive examination of the impacts of water regimes and fertilisation modalities on the soil bacterial community. The correlation between the observed findings and the extant body of literature is firmly established.
Response: Thank you for your appreciation
- The investigation of the effects of water deficit on the variety of the soil bacterial population is highly informative. Nevertheless, exploring plausible explanations for the apparent absence of substantial impact from water scarcity in this research is advisable, particularly when compared to previous scholarly works.
Response: We have considered these recommendations in the revised version of the manuscripts.
- The presentation of the discussion on many genera, including Bacillus, Acidobacteria, and Proteobacteria, under diverse situations is commendable. Nevertheless, it is worth exploring in more depth the potential ecological consequences of the reported alterations in these species on soil health and the nutrient cycling process.
Response: Our current data does not allow us to present an in-depth discussion of the ecological consequences, but we will consider these recommendations in our future research.
- The full examination of nitrifying bacteria and their correlation with greenhouse gas emissions is presented. Please provide further details on how these findings enhance our comprehension of the nitrogen cycle and emissions within agricultural systems.
Response: We considered these recommendations in the revised version of the manuscripts.
- It is important to highlight the congruity or disparity between the present findings and the extant body of literature. Examine any inconsistencies and possible factors contributing to them. This would offer a more comprehensive viewpoint of the impact of this work on wider scientific knowledge.
Response: Thank you for your suggestion. The different improvements of the introduction and discussion sections through the revision led to highlighting the congruity or disparity between the present findings and the extant body of literature.
Moderate editing of the English language required
Response: Thank you very much for your comments and suggestions. The English was crosschecked and improved as suggested.
References
Han, Q., Fu, Y., Qiu, R., Ning, H., Liu, H., Li, C., Gao, Y., 2023. Carbon Amendments Shape the Bacterial Community Structure in Salinized Farmland Soil. Microbiology Spectrum 11, e01012-01022.
Qiu, L., Zhang, Q., Zhu, H., Reich, P.B., Banerjee, S., van der Heijden, M.G., Sadowsky, M.J., Ishii, S., Jia, X., Shao, M., 2021. Erosion reduces soil microbial diversity, network complexity and multifunctionality. The ISME Journal 15, 2474-2489.
Yu, Z., Hu, X., Wei, D., Liu, J., Zhou, B., Jin, J., Liu, X., Wang, G., 2019. Long-term inorganic fertilizer use influences bacterial communities in Mollisols of Northeast China based on high-throughput sequencing and network analyses. Archives of Agronomy and Soil Science 65, 1331-1340.
Zhang, K., Maltais-Landry, G., James, M., Mendez, V., Wright, D., George, S., Liao, H.-L., 2022. Absolute microbiome profiling highlights the links among microbial stability, soil health, and crop productivity under long-term sod-based rotation. Biology and Fertility of Soils 58, 883-901.
